# Urban walkability through different lenses: A comparative study of GPT-4o and human perceptions

Musab Wedyan[1], Yu-Chen Yeh[2], Fatemeh Saeidi-Rizi[1]*, Tai-Quan Peng[3], Chun-Yen Chang[4]

1 School of Planning, Design and Construction, Michigan State University, East Lansing, Michigan, United States of America, 2 Department of Horticulture and Landscape Architecture, National Taiwan University, Taipei City, Taiwan, Taiwan, 3 Department of Communication, College of Communication arts and science, Michigan State University, East Lansing, Michigan, United States of America, 4 Department of Horticulture and Landscape Architecture, National Taiwan University, Taipei City, Taiwan, Taiwan

* saeidiri@msu.edu

## Abstract

Urban environments significantly shape our well-being, behavior, and overall quality of life. Assessing urban environments, particularly walkability, has traditionally relied on computer vision and machine learning algorithms. However, these approaches often fail to capture the subjective and emotional dimensions of walkability, due to their limited ability to integrate human-centered perceptions and contextual understanding. Recently, large language models (LLMs) have gained traction for their ability to process and analyze unstructured data. With the increasing reliance on LLMs in urban studies, it is essential to critically evaluate their potential to accurately capture human perceptions of walkability and contribute to the design of more pedestrian-friendly environments. Therefore, a critical question arises: can large language models (LLMs), such as GPT-4o, accurately reflect human perceptions of urban environments? This study aims to address this question by comparing GPT-4o's evaluations of visual urban scenes with human perceptions, specifically in the context of urban walkability. The research involved human participants and GPT-4o evaluating street-level images based on key dimensions of walkability, including overall walkability, feasibility, accessibility, safety, comfort, and liveliness. To analyze the data, text mining techniques were employed, examining keyword frequency, coherence scores, and similarity indices between the participants and GPT-4o-generated responses. The findings revealed that GPT-4o and participants aligned in their evaluations of overall walkability, feasibility, accessibility, and safety. In contrast, notable differences emerged in the assessment of comfort and liveliness. Human participants demonstrated broader thematic diversity and addressed a wider range of topics, whereas GPT-4o had more focused and cohesive responses, particularly in relation to comfort and safety. In addition, similarity scores between GPT-4o and the responses of participants indicated a moderate level of alignment between GPT-4o's reasoning and

**Data availability statement:** All relevant data are within the manuscript and its Supporting Information files.

**Funding:** The author(s) received no specific funding for this work.

**Competing interests:** The authors declare that there are no conflicts of interest associated with this study.

human judgments. The study concludes that human input remains essential for fully capturing human-centered evaluations of walkability. Furthermore, it underscores the importance of refining LLMs to better align with human perceptions in future walkability studies.

---

## 1. Introduction

Walkability has garnered considerable attention across various disciplines such as urban planning, public health, and transportation [1–3]. The quality of the walking environment is recognized also as a crucial component in enhancing community development [4], better human experience in historical sites [5], and reducing carbon emissions [6, 7].

Previous studies have constructed models to measure the perceived walkability such as using panoramic street view images and virtual reality [8–10]. Additionally, machine learning techniques, such as ResNet, have been employed to objectively quantify walkability based on pedestrian visual perception [11]. In addition, researchers applied deep learning algorithms to create a walkability index using micro and macro urban features [12–14]. Collectively, the extensive application of street view imagery and deep learning algorithms have enabled the development of methods to assess pedestrian walkability.

Recently, along with computer vision techniques in urban studies, large language models (LLMs) have become increasingly capable of performing a wide range of tasks, including text completion, sentiment analysis [15,16], and cross-language translation [17]. LLMs have also found applications in social science research, where they simulate human responses to survey questions on attitudes and behaviors [18,19]. The release of ChatGPT at the end of 2022 brought global attention [20–22]. Building on this momentum, the newly introduced GPT-4o model, with its multimodal capabilities, has further expanded the possibilities. For example, it has been applied to medical data [23–26], fake news detection [27], education [28, 29], business [30, 31], agriculture [32] and social science [33]. Those studies show that LLMs have been applied in different domains. However, despite that generative methods in the field of walkability are expected to grow [34], and the use will be expanding in urban tasks [35], the application of LLMs in the urban domain remains is still limited.

Overall, according to the literature, previous research has extensively utilized street view imagery and computer vision techniques to assess the physical attributes of walkable environments. However, the potential of studying the performance of LLMs in the urban walkability field is unexplored. Addressing this gap, we only aimed to explore the alignment of the human perspective of the perceived walkability and GPT-4o as one of the LLMs. We will answer the following questions: how well do these models accurately capture real-world human experiences of visual appeal? We examined the capabilities of GPT-4o in assessing the visual perception of walkability in urban areas by having it evaluate overall walkability, feasibility, accessibility, safety, comfort, and liveliness. By comparing paired images, we assessed their

ratings, text responses, and sentiment scores against those of human participants. Our findings highlight the limitations of GPT-4o in accurately perceiving urban environments and point to opportunities for refining LLM models to better align with human perspectives.

## 2. Literature review

### 2.1. Walkability perception

Walkability is increasingly acknowledged as a key element in promoting healthy communities [36], as well as enhancing social interaction and economic vitality within neighborhoods [37]. Walkability is typically characterized as the degree to which a built environment is accessible and appealing to individuals [38], whether they walk out of necessity, preference, or social engagement [39]. It also refers to individuals' perceptions of a street as a suitable place for walking [40]. As a subjective measure, walkability reflects the perceived quality of the environment and is shaped by personal assessments of its suitability for walking, making it challenging to quantify and assess objectively [41]. Collectively, these elements shape what is often referred to as perceived walkability [42,43].

The investigation of perceived walkability through subjective assessments has emerged as an effective approach to deepen our comprehension of the walking environment [44,45]. Among the various factors influencing perceived walkability, the concept of visual variety has emerged as a critical determinant of pedestrian satisfaction. Visual variety captures the richness and diversity of urban design elements that engage and attract pedestrians, enhancing the overall appeal of space [46]. Based on Maslow's hierarchy of needs, perceived walkability is suggested to consist of five dimensions: feasibility, accessibility, safety, comfort, and pleasurability [46]. Researchers commonly evaluate perceived walkability through four key dimensions: comfort, safety, utility, and appeal [46,47]. Other studies have referred to visual variety using terms such as imageability, complexity, transparency [40], and positive sensory experiences [48]. Building on this foundation, walkability has been systematically evaluated using five visual indicators established in early research: feasibility, accessibility, safety, comfort, and pleasurability [46].

According to [46], collectively, these indicators form the six categories of Visual Walkability Perception, providing a comprehensive framework for determining whether an environment visually supports walking. The visual walkability indicator provides an overall assessment of whether a location visually supports walking. Feasibility refers to factors that encourage walking, influenced by land use types and the diversity of available facilities. Accessibility addresses visible obstacles, such as dead-end streets or restricted access areas. Safety assesses a street's security based on crime, traffic accidents, and visual cues like graffiti, litter, and neglected buildings. Comfort examines how the street environment enhances the pedestrian experience, factoring in elements like street furniture, sidewalk width, urban design features, and accessibility facilities. Finally, pleasurability assesses the appeal of public spaces, reflecting how diverse, lively, enjoyable, and interesting they are for walking.

Yet walkability remains inherently subjective and challenging to quantify. In a recent review of the trends of walkability over time, it was concluded that methods for measuring walkability have shifted dramatically [49]. Early studies largely relied on measurement-based methods such as GIS-based assessments, and physical and image audits [50–56], as well as mixed-method approaches [57]. However, recent years have seen a growing emphasis on micro-level, street-based evaluations and SVI [58, 59] with applications including measuring psychological greenery and visual crowdedness [60] and object importance [61].

Although analyzing SVI by using CNN-based approaches has been effective in identifying physical features [62], they come with notable limitations. SVI and CNN do not inherently integrate textual opinions or subjective feedback from participants, which are crucial for understanding perceived walkability attributes like safety, aesthetics, or comfort [63,64]. Research has also highlighted the importance of textual opinions, such as those gathered from social media or surveys, to complement SVI in capturing the emotional and subjective dimensions of walkability [65,66]. This lack of interpretability makes it challenging for urban planners and policymakers to grasp the "reasoning" behind a algorithm's assessment.

Recently, Unlike CNNs, which focus on recognizing visual features, LLMs can analyze both text and images, allowing them to interpret the broader context surrounding an environment [35]. By leveraging both visual and textual data, LLMs bridge the gap between physical environment analysis and subjective user experiences. For example, LLMs can integrate data from computer vision analyses of street view imagery with textual user feedback, enabling a comprehensive under-standing of urban spaces [67]. This multimodal approach not only addresses the limitations of single-modality methods but also provides richer, more actionable insights for creating walkable cities.

### 2.2. ChatGPT advancements and uses

Recently, GPT-4o showcased extraordinary multimodal capabilities in decoding visual and textual content [68]. It has shown precision in functions such as detecting and identifying visual elements [69]. Another study explored the performance of GPT-4o in image classification by integrating images with textual descriptions. It was demonstrated that such a combination can significantly improve the accuracy of classification [70]. In addition, GPT-4o can identify and rank the perceived risk in traffic scenarios in images to some extent, despite that its evaluations do not always align with human judgment [71].

ChatGPT has been attracting attention from individuals across different backgrounds such as healthcare, and academia [20–22,72]. In the medical field, GPT-4 has played roles in enhancing radiology report assessments [73], conducting reviews on the digital twin concept [74], undertaking medical writing tasks [75], and aiding in licensing exams [76], medical education [77, 78], and visualizing internal body structures for diagnosis, research. Additionally, in a recent overview summarizing GPT's application across mathematics [79], physics [80–82], and communication [83].

Despite the wide use of LLM in different domains, there is a significant gap in understanding how these models perceive and evaluate outdoor environments compared to human perspectives. While previous studies have extensively explored human perceptions of urban environments through traditional methodologies such as surveys, mixed-methods approach, and deep learning, it is important and methodologically desirable to compare GPT-4o perceptions of outdoor environments in terms of visual walkability with human perceptions. This research is the first to investigate the alignment or divergence between LLM-based evaluations and human perceptions of walkability. This study seeks to evaluate the potential of Large Language Models (LLMs) as reliable tools for assessing environmental factors influencing human experiences, such as walkability, an area which has not been systematically compared in prior research. Based on this, the research addresses the following questions: how do the keyword frequencies in GPT-4o-generated descriptions of paired images compare to those in human-generated descriptions? In what ways do the sentiment scores of GPT-4o-generated descriptions differ from those of human-generated descriptions? Additionally, how do the coherence and similarity indices of GPT-4o-generated responses compare to those of human responses?

## 3. Methodology

Fig 1 presents a detailed overview of the research process across three distinct phases. The first phase involves the evaluation of paired images by both human participants and GPT-4o. The second phase consists of two scenarios: one where both GPT-4o and human participants choose the first image as having a higher rating, and another where both select the second image. After aligning the responses based on the selected image, the third phase synthesizes the findings by analyzing response coherence and keywords to identify themes more highly rated by either GPT-4o or human participants. This phase also compares sentiment scores across responses, performs LDA topic modeling, and analyzes keyword frequencies.

### 3.1. Data collection

  **3.1.1. Image selection and questionnaire structure.** Images were collected from Lansing, East Lansing, and Williamston, Michigan, using a horizontally held iPhone 14 for consistency. The selected images for this study aim to represent multiple perspectives of the urban environment, encompassing a broad spectrum of developments in chosen

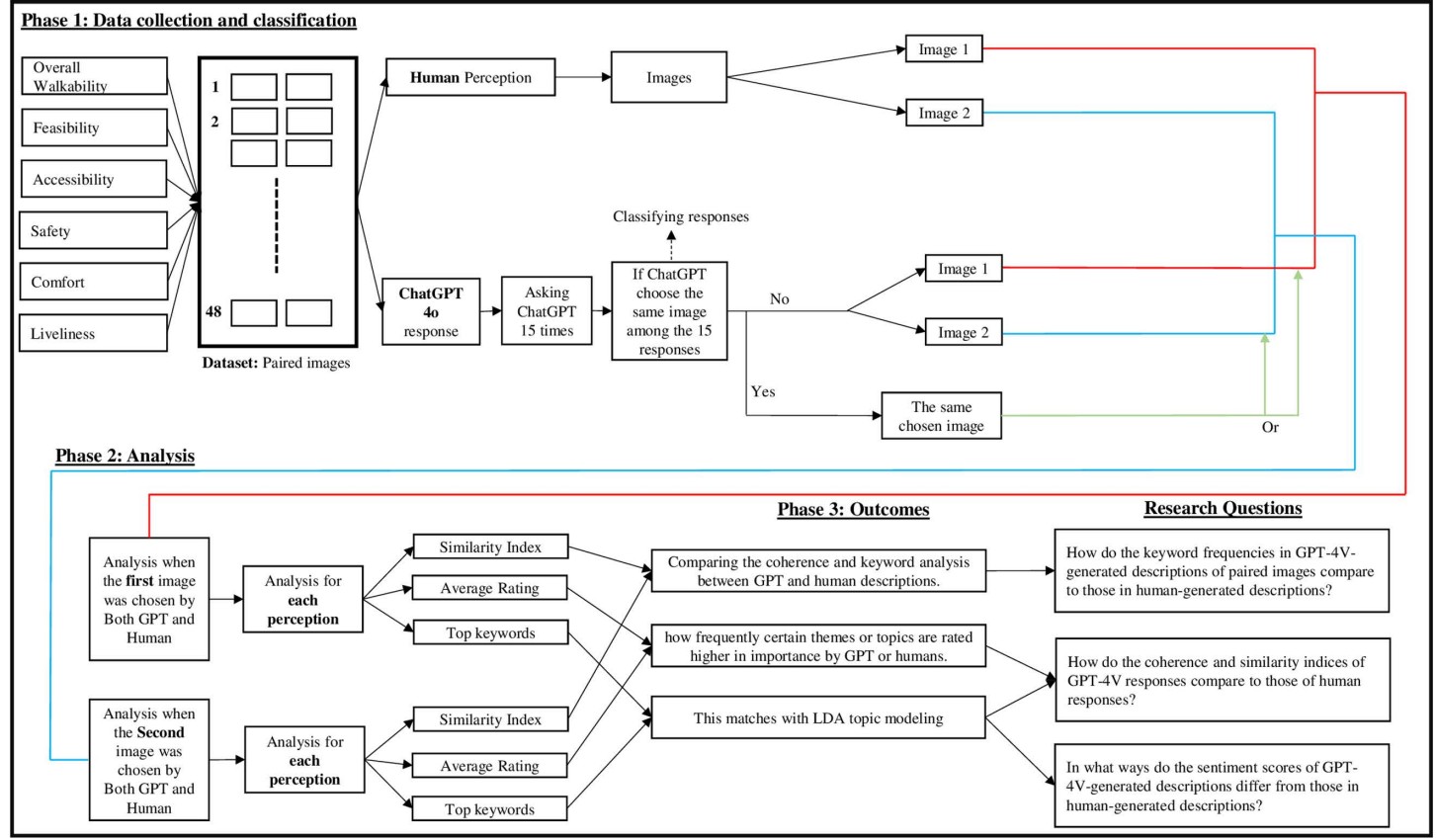

**Fig 1. Research Methodology.**

areas to illustrate the diversity present in city landscapes. The evaluation of these variations was based on the subjective evaluation of the authors. The selection was designed to represent diverse perspectives on urban development, covering areas with various levels of greenery, pavement conditions, population density, vehicle and pedestrian flow, and spatial openness or constriction. This variation included both tree-lined streets and concrete-dominated areas, as well as crowded streets and sparsely populated spaces, reflecting a wide range of urban conditions. These diverse images provided a holistic view of walkability, as influenced by both natural elements and urban infrastructure, capturing how these factors impact perceptions of walkability for both human participants and GPT-4o.

Each image was assigned a unique identifier ranging from 1 to 106 to facilitate randomization and organization. The randomization process was conducted using a Python script, which paired the images to create sets for comparison. This approach was employed to avoid any bias in the selection process and to ensure a fair representation of diverse urban environments. Using this approach, some images were excluded from the final survey for distinct reasons. Images 25, 32, 58, 59, 72, 84, 88, 96, and 102 were not selected during randomization. This resulted in a final selection of 48 unique image pairs that were included in the survey. The finalized image set was designed to provide a comprehensive range of urban conditions, capturing both natural and built features that influence walkability perceptions. The systematic pairing ensured that participants evaluated images reflecting real-world variability in urban design, allowing for robust comparisons of human and GPT-4o perceptions of walkability. Table 1 summarizes the number of respondents and image pairs which were included in the survey and subsequent analysis in the "images numbers" column.

**Table 1. Number of respondents and image pairs used in different groups of human survey.**

| Groups | Number of Respondents | Image Pairs Used | | | | | |
|---|---|---|---|---|---|---|---|
| | | Overall Walkability | Feasibility | Accessibility | Safety | Comfort | Liveliness |
| 1 | 38 | 44,1 | 60,56 | 53,100 | 64,3 | 103,87 | 8,67 |
| 2 | 26 | 77,19 | 105,26 | 75,83 | 35,17 | 95,101 | 41,81 |
| 3 | 13 | 28,94 | 98,14 | 30,2 | 42,91 | 11,43 | 15,90 |
| 4 | 11 | 48,12 | 65,68 | 66,50 | 86,104 | 73,9 | 18,20 |
| 5 | 39 | 27,45 | 23,16 | 46,51 | 34,52 | 49,62 | 61,70 |
| 6 | 22 | 69,54 | 37,21 | 22,57 | 85,63 | 4,71 | 36,33 |
| 7 | 13 | 24,97 | 40,82 | 10,89 | 31,79 | 13,99 | 106,47 |
| 8 | 6 | 6,39 | 74,80 | 38,55 | 29,92 | 76,93 | 78,5 |

Participants in the main survey compared pairs of images of urban settings based on six aspects of walkability. These aspects included overall walkability (ease and attractiveness of walking), feasibility (practicality of walking based on individual and environmental factors), accessibility (how well the area accommodates diverse abilities), safety (perceived security), comfort (pedestrian comfort level), and liveliness (vibrancy of the area). Before conducting the primary survey, a pilot study was carried out to assess the survey design and definitions of walkability. Participants understood the six key aspects of walkability—overall walkability, feasibility, accessibility, safety, comfort, and liveliness, but minor adjustments were made to improve clarity and ensure consistent interpretation. The initial survey was shortened for the main study due to its length. Finally, the survey, consisting of 12 question sets, took about 15–20 minutes to complete, and participants were instructed to respond without using AI tools to ensure authentic responses. S1 Table shows the responses of the human participants for paired images. This study was conducted in compliance with the ethical standards set forth by the Institutional Review Board (IRB) of Michigan State University (MSU). Approval for the study was granted under the protocol MSU Study ID: STUDY00010749, covering the period from May 13, 2024, to July 14, 2024. All participants provided written informed consent before participating in the study. The consent process adhered to the guidelines approved by the MSU IRB to ensure participants were fully informed about the objectives of the study, procedures, and their rights, including the option to withdraw at any time. After obtaining IRB approval, minor modifications were made to the study protocol. The initial survey was revised to shorten its length based on feedback from a pilot study to improve participant engagement. These modifications did not alter the core research objectives and remained within the scope of the original IRB approval.

The participants for this study were randomly selected through an online survey platform, which was distributed to a wide audience to ensure diversity in responses. The recruitment process did not target specific age groups, professions, or cultural backgrounds, allowing for a broad participant pool. This random selection approach helps mitigate biases in recruitment and enhances the generalization of the findings. While specific demographic targeting was not employed, the survey yielded 174 responses from individuals with diverse characteristics, including a range of ages (from 18 to 65± years), professions (e.g., students, healthcare workers, educators, and urban planning professionals), and cultural backgrounds, as detailed in section 4.1.

**3.1.2. ChatGPT prompting.** Various strategies have been employed to enhance the output of GPT-4o such as the use of composite images [84], comparing images in pairs [85], or employing multimodal cooperation [86]. Another technique is converting visual information into text using prompts like "What's in this image?". This method shows significant potential, especially when processing large volumes of images that appear in a temporal sequence [87]. However, minor variations in prompts can lead to inconsistent outputs [88]. To address this, methods like self-consistency or bootstrapping involve re-prompting multiple times with different text permutations and averaging the results, improving overall accuracy [89–91].

This method involves repeating the prompting process multiple times, each with a different permutation of the text, and then extracting the mean output. This aggregated output typically achieves higher accuracy than a single prompt.

In our study, we ensured consistency by utilizing the same pairs of images in both the surveys and the GPT-4o web interface. For example, the image pair of 44,1 was prompted to GPT 15 times and GPT-4o and prompted to provide their evaluation of overall walkability. These image pairs were uploaded as composite images into GPT-4o, and self-consistency techniques were used to improve the reliability of the model's output. By aligning the image pairs used in both the surveys and the GPT-4o web interface, we ensured that the results were directly comparable. The individual images had a size of 4023*3024 pixels. GPT-4o considers the image on the left as the first image while the image on the right is the second one. We prompted GPT-4o between 15 July and 1 August 2024. Each prompt was in a new chat window by using temporary chat in GPT-4o. The use of temporary chat windows was crucial to ensure that each prompt was processed independently, avoiding any potential carryover effects or contextual memory from previous interactions. This approach minimized bias and ensured that the model's output for each prompt was generated without influence from earlier conversations, thereby enhancing the consistency and reliability of the results. In each prompt, we requested GPT-4o to rank the images from 1–10 and describe the perception of each walkability perception. For example, when asking about overall walkability, we wrote the following prompt: "How do you rate the Walkability of this environment for both photos from 1–10? Based on the photo you rated higher, why do you think it is more Walkable? (Please explain your opinion in at least 20 words). Overall Walkability: This measures the ease and appeal of walking around the area shown in the image.

## 4. Results

### 4.1. Demographic variables of respondents in the survey

The survey included 174 participants, with a balanced gender representation: 47% female, 50% male, and 2% preferring not to disclose their gender. The sample primarily consisted of younger adults, with 34% aged 18–25 and 33% aged 26–35, while participation decreased in older age groups (15% aged 36–45, 7% aged 46–55, 6% aged 56–65, and 1% over 65). Geographically, the majority of participants resided in urban areas (87%) compared to rural areas (12%). Most respondents were from the United States (61%), followed by Taiwan (15%), with smaller contributions from Jordan, Canada, China, Germany, and other countries (each 4% or less). Walking habits varied among participants: 22% reported walking daily, 27% walked four to five times a week, and 25% walked two to three times a week, while 12% walked once a week and 12% rarely walked. In terms of weekly walking duration, 52% walked for 30 minutes to 1 hour, 39% for less than 30 minutes, and 8% for 1–2 hours, while only 0.5% walked for more than 2 hours.

To maintain the integrity and consistency of the analysis, responses were filtered according to six specified dimensions of walkability: overall walkability, feasibility, accessibility, safety, comfort, and liveliness. Exclusions were made based on predetermined criteria: responses were deemed "Equal" if participants assigned the same ratings to both images, "Invalid" if they were incomplete or lacked significant differentiation, or inconsistent for particular images. Notably, this exclusion process was conducted at the level of individual responses for specific images, rather than dismissing entire participants, thereby preserving valid responses for other perceptions from the same individuals. Following this rigorous filtering, the final counts of analyzed responses for each dimension were 168 for overall walkability, 164 for feasibility, 162 for accessibility, 161 for safety, 165 for comfort, and 162 for liveliness. This selective methodology ensured maximizing the contributions of participants, thus facilitating a thorough comparison of human and GPT-4o of all perceptions. The total number of responses from participants for each perception included in the analysis is shown in Table 2.

### 4.2. Consistency of GPT-4o responses

GPT-4o's responses exhibited two distinct patterns: it either consistently chose the same image across all prompts or varied its selection between the two images. GPT-4o's responses displayed a clear answering pattern: for 41 out of the 48 image pairs, it consistently selected the same image (either the first or the second) across all prompts. In the remaining 7

**Table 2. Number of responses of GPT-4o and participants' responses for each perception.**

| | Decision | Perception | | | | | |
|---|---|---|---|---|---|---|---|
| | | Overall walkability | Feasibility | Accessibility | Safety | Comfort | Liveliness |
| Participants | First image | 68 (39%) | 69 (40%) | 70 (40%) | 74 (43%) | 109 (62%) | 73 (42%) |
| | Second image | 100 (57%) | 95 (55%) | 92 (52%) | 87 (50%) | 56 (33%) | 89 (51%) |
| | Equal | 2 (1%) | 4 (2%) | 2 (1%) | 3 (1%) | 3 (2%) | 9 (5%) |
| | Invalid | 4 (3%) | 6 (3%) | 10 (7%) | 10 (6%) | 6 (3%) | 3 (2%) |
| Total responses included in the analysis | | 168 | 164 | 162 | 161 | 165 | 162 |
| Total responses | | **174** | | | | | |
| GPT-4o | First image | 45 (38%) | 49 (41%) | 53 (44%) | 45 (38%) | 56 (47%) | 45 (38%) |
| | Second image | 75 (62%) | 71 (59%) | 67 (56%) | 75 (62%) | 64 (53%) | 75 (62%) |
| | Equal | 0 (0%) | 0 (0%) | 0 (0%) | 0 (0%) | 0 (0%) | 0 (0%) |
| | Invalid | 0 (0%) | 0 (0%) | 0 (0%) | 0 (0%) | 0 (0%) | 0 (0%) |
| Total responses | | **120** | | | | | |

pairs, GPT-4o alternated between selecting the first and second images. To determine the number of chosen responses for analysis, we calculated the similarity index for the generated responses out of the 15 responses. S2 Table shows what image GPT-4o chose as a higher rank, the number of chosen responses out of 15. For example, under the feasibility category, when the image numbers (60,56), (65,68), and (23,16), the second image was consistently chosen.

The similarity index was calculated by comparing responses, with the first response serving as the baseline reference. Pairwise comparisons were conducted across all 15 responses, starting with the first response compared to the second, followed by the first compared to the third, and so forth, until all responses had been evaluated. This method enabled an assessment of the cumulative consistency of the responses as additional outputs were generated. The same approach was applied uniformly to all responses, up to the final response (response number 15). (S1–S6 Fig) in the supporting information present the similarity index for each perception of walkability relative to the number of generated responses by GPT-4o. Although some fluctuation in the similarity index was observed, the values remained within a relatively narrow range. Based on this stability, 15 responses were selected for further analysis.

### 4.3. Alignment between GPT-4o and human responses

To assess the alignment between the responses from GPT-4o and those from human participants, we matched the responses when GPT-4o and participants chose either image 1 or image 2 in each pair. The number of pair image sets that were selected when the first image was chosen was 21 pairs while when the second image was chosen, we had 29 pairs. The number of participants based on the chosen images is illustrated in Table 2. The results highlighted notable differences in image selection across various perceptual categories. In terms of overall walkability, feasibility, accessibility, and safety, both participants and GPT-4o demonstrated a tendency to select the second image, with participants choosing it 57%, 55%, 52%, and 50% of the time, respectively, while GPT-4o showed comparable preferences at 62%, 59%, 56%, and 62%. However, a key divergence was observed in the comfort perception, where participants significantly favored the first image (62%), while GPT-4o leaned towards the second image (53%). A similar discrepancy emerged in the liveliness perception. Additionally, participant responses displayed more variability, with some instances of "equal" or "invalid" responses, the latter referring to cases where participants provided identical answers across multiple questions.

## 4.4. Comparing GPT-4o and human ratings

We compared the ratings between GPT-4o and human participants for two sets of images: Image 1 and Image 2. Independent samples t-tests were conducted to evaluate whether there were significant differences between the ratings assigned by humans and the GPT-4o model. For Image 1, the mean rating given by participants was (M = 7.80) with a standard deviation of (SD = 0.60), while the GPT-4o model assigned a mean rating of (M = 7.42) with a standard deviation of (SD = 0.70). The analysis did not reveal a statistically significant difference between the ratings of humans and the GPT-4o model, (t(N) = 1.91), (p = .063), suggesting that the GPT-4o model's ratings were generally similar to those of human participants for this set of images. For Image 2, participants rated the images with a mean of (M = 7.65) and a standard deviation of (SD = 0.81), whereas the GPT-4o model provided ratings with a mean of (M = 7.76) and a standard deviation of (SD = 0.47). The t-test indicated no significant difference between the two groups, (t(N) = -0.64), (p = .526), implying that the ratings assigned by the GPT-4o model were not substantially different from those of the human participants in this case.

## 4.5. Content analysis

### 4.5.1. Similarity index between GPT-4o and participants' responses.
We assessed the alignment between the textual responses of GPT-4o and human participants by examining the average similarity index to compare their reasoning and descriptive alignment. This was applied to all responses regardless of the perception. Text preprocessing steps, such as tokenization and stop-word removal, were applied before vectorization to ensure meaningful comparisons. This approach follows methodologies established in previous studies on text similarity [92].

The textual data was then vectorized, and cosine similarity was used to measure alignment between GPT-4o and human responses. Cosine similarity is a widely accepted metric for quantifying textual similarity by comparing vectorized representations of text, as discussed in [93, 94]. The threshold value for cosine similarity varies depending on the context, dataset, and application. It is often empirically determined or adaptively set to meet the needs of specific tasks, such as clustering, text classification, or similarity searches. Typical thresholds for high similarity range from 0.7 to 0.9, with values closer to 1.0 indicating stronger similarity, particularly in normalized datasets [95,96]. In the context of this study, a score above 0.4 is generally interpreted as moderate alignment, while scores closer to 0.6 or higher suggest stronger alignment.

The findings revealed a moderate degree of alignment, with an average similarity score of 0.4575 when the first image was chosen and 0.4615 for the second image. While this indicates that GPT-4o is capable of partially mimicking human decision-making processes, the results suggest that there are notable differences in how the two approach visual tasks, particularly in terms of depth and nuance.

### 4.5.2. Topic modeling.
We compared the responses from GPT-4o and human participants by looking into the topic modeling results and coherence scores across six categories of perception. Fig 2 shows the number of topics when the first image was selected by both groups. Human participants identified nine topics relating to overall walkability perception, with a coherence score of 0.358. In comparison, GPT-4o generated three topics, but with a slightly lower coherence score of 0.317. Regarding feasibility perception, humans produced five topics with a coherence score of 0.360, whereas GPT-4o identified only two topics, which resulted in a coherence score of 0.283. When it comes to accessibility perception, human participants identified eight distinct topics, achieving a coherence score of 0.392, while GPT-4o generated four topics with a score of 0.351. In terms of safety perception, both GPT-4o and human participants had similar outcomes. Human participants generated four topics with a coherence score of 0.377, whereas GPT-4o identified three topics and attained a slightly lower coherence score of 0.368. Interestingly, GPT-4o excelled in the comfort perception. Although it identified fewer topics (three), it achieved a higher coherence score of 0.378 compared to the four topics and a 0.356 score produced by human participants. For liveliness perception, human participants identified two topics with a notably higher coherence score of 0.458, while GPT-4o generated five topics, but with a lower score of 0.317.

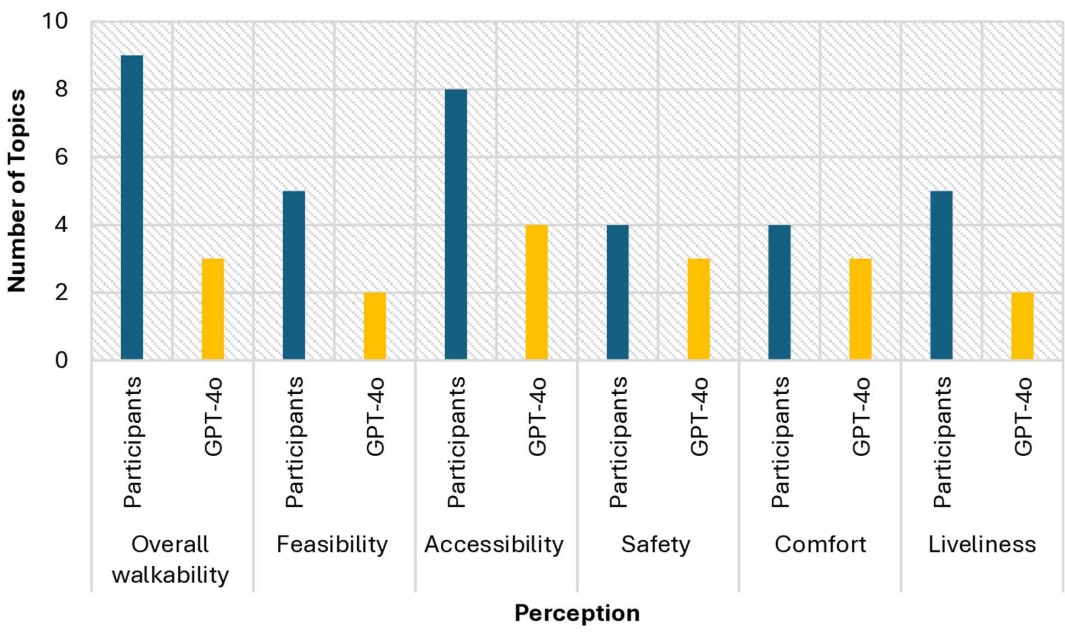

**Fig 2. Number of Topics (First Image).**

Fig 3 shows the number of topics when the second image was chosen. Both GPT-4o and human participants identified three topics related to walkability, but human responses were more cohesive, achieving a coherence score of 0.363 compared to GPT-4o's 0.332. In assessing feasibility, humans identified five topics, resulting in a higher coherence score of 0.398, while GPT-4o found three topics with a coherence score of 0.348. For accessibility, human participants were able to identify nine topics with a coherence score of 0.363, while GPT-4o only generated four topics, resulting in a much lower score of 0.277. This suggests that human responses captured a broader spectrum of accessibility-related themes in a more cohesive manner. For safety, GPT-4o's coherence score was considerably lower (0.285) compared to the human participants' score of 0.387, indicating that human responses were more coherent and comprehensive. In the comfort category, humans identified ten topics with a coherence score of 0.355, whereas GPT-4o identified seven topics with a slightly lower coherence score of 0.333. Finally, for liveliness, human participants identified ten topics with a coherence score of 0.391, whereas GPT-4o only identified two topics, resulting in a coherence score of 0.297.

**4.5.3. Top keywords.** When analyzing the responses of the first image, notable differences in the top keywords between human participants and GPT-4o reveal distinct approaches to perceiving outdoor spaces. For example, human responses frequently included experiential and descriptive words like "trees," "shade," "people," and "comfortable," reflecting a focus on the sensory experience and aesthetic quality of the environment. Humans often described how space made them feel and how specific natural elements contributed to comfort and liveliness. Words such as "traffic" and "obstacles" further indicated that humans were concerned with practical aspects of safety. In contrast, GPT-4o focused on more structured and functional terms like "pedestrian," "path," "mobility," and "amenities." This language suggests that GPT-4o approached the image from a more technical perspective, emphasizing the design and infrastructure of the outdoor space, such as the presence of sidewalks, accessibility features, and pedestrian pathways. GPT-4o's responses were centered around how space functions for movement and public use, with less attention to subjective feelings or sensory details. In the second image, the differences between human and GPT-4o responses became even more pronounced. Humans continued to focus on the visual and sensory aspects of the environment, with frequent use of terms like "green," "grass," "quiet," and "lively." These keywords highlight the participants' attention to natural features and

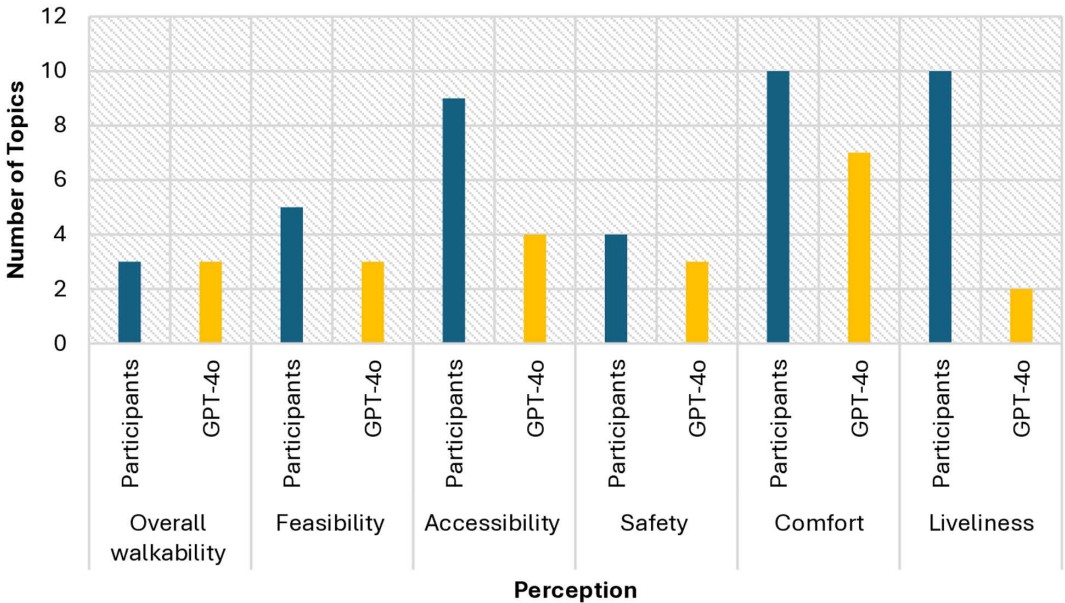

**Fig 3. Number of Topics (Second Image).**

the ambience of space, indicating a more holistic perception that integrates the physical appearance of the area with its emotional impact. Human participants evaluated space based on how peaceful or active it seemed, using language that suggested an assessment of the overall vibrancy and aesthetic appeal. On the other hand, GPT-4o's responses were dominated by keywords such as "urban," "area," "street," and "mobility," once again emphasizing the functional design of the space. While humans highlighted specific natural features and the emotional atmosphere, GPT-4V remained more objective, concentrating on infrastructure and the practical usability of the environment.

## 5. Discussion

Prior research has shown that machine learning and computer vision methods are effective in analyzing image datasets, including those from Google Street View, to predict various factors such as scene complexity, safety, and socioeconomic conditions [64,97,98]. The emergence of vision-language models, such as ChatGPT [99], PaLM [100], and LLaMa [101] offer new opportunities for evaluating images in more comprehensive ways. Despite their promise, the extent to which these models align with human perception, particularly in urban walkability assessments, remains insufficiently explored. Building on this foundation, our study explored GPT-4o's potential as a tool for assessing visual walkability in urban environments by comparing its evaluations with those of human participants. Our findings reveal a strong alignment between GPT-4o and human participants in assessing overall walkability, feasibility, accessibility, and safety. However, notable differences emerged in the assessment of comfort and liveliness, where human participants provided more thematically diverse insights, while GPT-4o's responses were more structured and cohesive, particularly regarding comfort and safety. Similarity scores suggest a moderate level of alignment between GPT-4o's reasoning and human judgments, highlighting its potential to systematically complement human evaluations of urban walkability with a systematic and focused approach. These findings contribute to ongoing discussions on AI's role in urban design and suggest promising avenues for integrating AI-driven assessments into urban planning processes.

First, the results in Table 2 indicated that GPT-4o and human participants match each other in the choice of images for perceptions such as general walkability, feasibility, accessibility, and safety. However, human responses had "equal"

answers when choosing some images, which indicates a nuance in human decision-making that was not replicated by GPT-4o. That would suggest that in areas where human preferences can be mimicked, the decision-making done by GPT-4o may not possess the same level of complexity and flexibility as that found in human judgments on most subjective perceptions.

Second, while the similarity scores—0.4575 for the first image and 0.4615 for the second indicated a moderate level of agreement, they also suggest that GPT-4o's responses do not perfectly mirror human judgments. The slight difference in scores shows that GPT-4o might align more closely with human participants when evaluating certain images, but its reasoning does not fully capture the nuances of human perception. This moderate alignment points to GPT-4o's potential to assist in tasks that require subjective judgment, such as urban design and walkability assessments, but it also highlights the need for caution when relying solely on AI models for decisions that involve complex human-centered evaluations.

Third, the comparison between GPT-4o and human participants shows that humans consistently identified a wider variety of topics across most perception categories, such as walkability, feasibility, and liveliness, although their responses varied in cohesiveness. In contrast, GPT-4o generated fewer topics but delivered more cohesive and focused responses, especially regarding comfort and safety. This indicates that while humans capture a broader range of themes, GPT-4o provides more structured and streamlined interpretations based on the perception being assessed. Notably, GPT-4o had difficulty with more complex and abstract perceptions like liveliness and accessibility, where it identified fewer topics and had lower coherence scores. On the other hand, human participants showed a more consistent ability to recognize a wide array of factors across different perceptions and images. Therefore, we argue that humans excel in identifying thematic diversity, while GPT-4o is skilled at organizing and simplifying more straightforward perceptions.

Fourth, regarding the top keywords, while humans emphasized specific natural features and the emotional atmosphere, GPT-4o remained more objective, concentrating on infrastructure and the practical usability of the environment. This aligns with earlier research on human perceptions of walkability in urban spaces. Studies such as those emphasized the importance of physical infrastructure like sidewalks and road conditions in shaping walkability perceptions [40,102]. In contrast, GPT-4o's broader focus on the usability of space presents a different approach, one that is more grounded in specific infrastructural details. This divergence mirrors the findings by [60], who argued that while visual elements are essential, the subjective nature of walkability makes it challenging to capture the full scope of human experience through automated tools alone. Our study reinforces this notion by demonstrating that while GPT-4o can provide a consistent overview of environmental quality, it often misses the context-specific insights that human evaluators offer. Therefore, integrating human perspectives to fully capture the depth of contextual and experiential details is needed to understand human emotions in urban spaces [103]. Further supporting the need for a balanced approach, by integrating GPT-4o with human judgment, urban planners can benefit from the strengths of both approaches, leading to more effective and responsive urban design solutions.

Overall, this study demonstrates the potential practical use of LLMs, such as GPT-4o, to complement human-centered urban planning by providing structured, scalable assessments of walkability. While GPT-4o showed notable alignment with human evaluations in aspects such as overall walkability and feasibility, its applications can extend beyond simple evaluations to play a more dynamic role in urban design processes. For instance, LLMs could be employed in automated urban audits, analyzing street-level imagery to identify infrastructure gaps such as the absence of pedestrian crossings, narrow sidewalks, or insufficient greenery. This capability could save significant time and resources, particularly for large-scale urban projects. Another promising application lies in scenario simulations, where planners could upload mock-up designs or proposed changes to urban spaces and receive AI-driven feedback on how these alterations might influence walkability indicators such as comfort, accessibility, or safety. Additionally, LLMs could enhance public engagement by acting as an intermediary in community participation initiatives. It can translate technical urban design elements into more accessible language for residents, helping stakeholders better understand proposed plans and prioritize elements that align with public sentiment.

Despite these promising applications, the limitations of GPT-4o must be acknowledged, particularly in addressing cultural and personal factors that heavily influence human judgments of walkability. Cultural norms shape perceptions of aspects such as safety, liveliness, and accessibility differently across regions. For instance, a vibrant urban space in one cultural context might be perceived as chaotic or unsafe in another. Similarly, personal preferences, including mobility needs or aesthetic values, add layers of subjectivity that GPT-4o struggles to capture without explicit input. These limitations stem from GPT-4o's reliance on text and image data, which, while powerful, cannot fully account for experiential or emotional connections to urban spaces. To address these subjective differences, future studies should incorporate more diverse datasets that reflect the cultural and geographic variability of urban spaces. Additionally, GPT-4o's capabilities could be enhanced through multimodal data integration, including pedestrian movement patterns, audio cues, and climate data, to better simulate human sensory experiences and contextualize walkability evaluations. However, even with these advancements, human-centered design decisions require nuanced, context-specific insights that LLMs cannot replicate. LLMs should therefore be seen as a tool to augment human expertise, particularly in subjective and culturally sensitive areas of urban planning, rather than as a replacement for human judgment. Incorporating studies that compare different urban typologies, such as high-density versus low-density areas, could provide a new understanding of the role of built environmental characteristics. Furthermore, exploring the application of LLMs-driven tools in collaborative design processes with stakeholders, including urban planners, architects, and community members, could bridge the gap between data-driven models and practical implementation. In addition, the results emphasize GPT-4o's proficiency in delivering structured and coherent assessments; however, they also reveal certain limitations, particularly regarding its ability to grasp more abstract and subjective metrics such as liveliness and accessibility. These perceptions are intricately linked to experiential, and contextual elements that may not be entirely quantifiable with the existing LLMs models and methodologies utilized in this research. Recognizing these limitations is essential to maintain realistic expectations for LLMs applications within urban studies. Additionally, the images used in this study were geographically taken in urban areas in Michigan, which may limit the applicability of the findings to other regions with distinct urban characteristics. The reliance on specific image types and the exclusion of certain human responses, such as biased or equal ratings, may have restricted the depth of the analysis. Given these limitations, future research should address these limitations to enhance the understanding of the role of LLMs in urban planning.

## 6. Conclusion

In conclusion, our results showed that while GPT-4o can help in measuring the perception of people of urban walkability, it cannot fully replicate the depth of human perception. The integration of LLMs into urban planning should be approached with caution, ensuring that these tools are used to augment, rather than replace, the perceptions of people. By refining LLM algorithms and incorporating human feedback, there is potential to develop more effective and responsive tools for urban analysis, ultimately leading to better design. However, this study has several limitations. The relatively small and demographic homogeneity sample may not fully capture the diversity of urban walkability perceptions across different populations. So, our main conclusion is that humans continue to have the final authority in decision-making. Instead of replacing urban planners, future research should concentrate on creating customized LLMs solutions for urban studies.

## Supporting information

**S1 Table.  Human participants' responses to paired images.**
(DOCX)

**S2 Table.  GPT-4V responses to paired images.**
(DOCX)

**S1 Fig. Similarity Index of GPT-4V Responses Regarding Overall Walkability Perception.**
(TIF)

**S2 Fig. Similarity Index of GPT-4V Responses Regarding Feasibility Perception.**
(TIF)

**S3 Fig. Similarity Index of GPT-4V Responses Regarding Accessibility Perception.**
(TIF)

**S4 Fig. Similarity Index of GPT-4V Responses Regarding Safety Perception.**
(TIF)

**S5 Fig. Similarity Index of GPT-4V Responses Regarding Comfort Perception.**
(TIF)

**S6 Fig. Similarity Index of GPT-4V Responses Regarding Liveliness Perception.**
(TIF)

## Acknowledgments

This study was enriched by collaborative discussions with members of the HealthScape Lab at Michigan State University. We are also deeply thankful to Tai-Quan Peng and Chun-Yen Chang for their insightful feedback and contributions during the review of the manuscript. We gratefully acknowledge Michigan State University for offering the infrastructure and resources necessary to carry out this research. Our heartfelt thanks also go to the editorial team and the anonymous reviewers for their valuable comments and suggestions on the earlier draft of this paper.

## Author contributions

**Conceptualization:** Musab Wedyan, Yu-Chen Yeh, Fatemeh Saeidi-Rizi.

**Data curation:** Musab Wedyan, Yu-Chen Yeh.

**Formal analysis:** Musab Wedyan, Tai-Quan Peng.

**Investigation:** Fatemeh Saeidi-Rizi.

**Methodology:** Musab Wedyan, Yu-Chen Yeh, Fatemeh Saeidi-Rizi, Tai-Quan Peng, Chun-Yen Chang.

**Resources:** Fatemeh Saeidi-Rizi.

**Software:** Musab Wedyan.

**Supervision:** Fatemeh Saeidi-Rizi.

**Visualization:** Musab Wedyan.

**Writing – original draft:** Musab Wedyan.

**Writing – review & editing:** Musab Wedyan, Fatemeh Saeidi-Rizi.

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
