## [Decision Letter · Decision Letter 0]

26 Dec 2024

PONE-D-24-54765Urban Walkability Through Different Lenses: A Comparative Study of GPT-4o and Human PerceptionsPLOS ONE

Dear Dr. Saeidi-Rizi,

Thank you for submitting your manuscript to PLOS ONE. After careful consideration, we feel that it has merit but does not fully meet PLOS ONE’s publication criteria as it currently stands. Therefore, we invite you to submit a revised version of the manuscript that addresses the points raised during the review process.

Note from associate editor

The manuscript offers valuable insights into AI and human perception regarding walkability. It attempts to provide an answer to the role of AI in preceding the built environment, similar to humans. 

The associate editor and the reviewer have provided comments below, which should be addressed carefully. 

Associate Editor's comments

1- The introduction needs to be restructured by addressing the gap in literature, research problem and research aim.   

2- The sample size of the survey should be clarified. Providing a detailed description of the sample size is essential. 

3- The titles end with punctuation marks such as full stops and semicolons. Such marks should be removed. 

4- The discussion section needs to add a couple of sentences that discuss the current findings from conducting a comparative study between Chat-4o and human responses. Additionally, the research limitations of utilising qualitative study should also be discussed. 

5- The conclusion section summarizes the central finding, but it is important to expand on suggestions for future research.    

6- It is essential to review the latest article published by PLOS One to ensure the continuity of the research direction.

7- For compliance with PLOS One requirements, it is essential to provide the approval for human subjects research from the Institutional Review Board (IRB) or an equivalent ethics committee at Michigan State University, where the authors are affiliated.

prior 

We look forward to receiving your revised manuscript.

Kind regards,

Prof. Abeer Elshater

Academic Editor

PLOS ONE

Journal Requirements:

7. Please include captions for your Supporting Information files at the end of your manuscript, and update any in-text citations to match accordingly. Please see our Supporting Information guidelines for more information: http://journals.plos.org/plosone/s/supporting-information .

8. We note that this data set consists of interview transcripts. Can you please confirm that all participants gave consent for interview transcript to be published?

If they DID provide consent for these transcripts to be published, please also confirm that the transcripts do not contain any potentially identifying information (or let us know if the participants consented to having their personal details published and made publicly available). We consider the following details to be identifying information:

- Names, nicknames, and initials

- Age more specific than round numbers

- GPS coordinates, physical addresses, IP addresses, email addresses

- Information in small sample sizes (e.g. 40 students from X class in X year at X university)

- Specific dates (e.g. visit dates, interview dates)

- ID numbers

Or, if the participants DID NOT provide consent for these transcripts to be published:

- Provide a de-identified version of the data or excerpts of interview responses

- Provide information regarding how these transcripts can be accessed by researchers who meet the criteria for access to confidential data, including:

a) the grounds for restriction

b) the name of the ethics committee, Institutional Review Board, or third-party organization that is imposing sharing restrictions on the data

c) a non-author, institutional point of contact that is able to field data access queries, in the interest of maintaining long-term data accessibility.

d) Any relevant data set names, URLs, DOIs, etc. that an independent researcher would need in order to request your minimal data set.

For further information on sharing data that contains sensitive participant information, please see: https://journals.plos.org/plosone/s/data-availability#loc-human-research-participant-data-and-other-sensitive-data

If there are ethical, legal, or third-party restrictions upon your dataset, you must provide all of the following details (https://journals.plos.org/plosone/s/data-availability#loc-acceptable-data-access-restrictions):

1. A complete description of the dataset

2. The nature of the restrictions upon the data (ethical, legal, or owned by a third party) and the reasoning behind them

3. The full name of the body imposing the restrictions upon your dataset (ethics committee, institution, data access committee, etc)

4. If the data are owned by a third party, confirmation of whether the authors received any special privileges in accessing the data that other researchers would not have

5. Direct, non-author contact information (preferably email) for the body imposing the restrictions upon the data, to which data access requests can be sent

Additional Editor Comments:

The manuscript offers valuable insights into AI and human perception regarding walkability. It attempts to provide an answer to the role of AI in preceding the built environment, similar to humans.

The editor and the reviewer have provided comments below, which should be addressed carefully.

Associate editor's comments

1- The introduction needs to be restructured by addressing the gap in literature, research problem and research aim.

2- The sample size of the survey should be clarified. Providing a detailed description of the sample size is essential.

3- The titles end with punctuation marks such as full stops and semicolons. Such marks should be removed.

4- The discussion section needs to add a couple of sentences that discuss the current findings from conducting a comparative study between Chat-4o and human responses. Additionally, the research limitations of utilising qualitative study should also be discussed.

5- The conclusion section summarizes the central finding, but it is important to expand on suggestions for future research.

6- It is essential to review the latest article published by PLOS One to ensure the continuity of the research direction.

7- For compliance with PLOS One requirements, it is essential to provide the approval for human subjects research from the Institutional Review Board (IRB) or an equivalent ethics committee at Michigan State University, where the authors are affiliated.

Reviewers' comments:

Reviewer's Responses to Questions

**Comments to the Author**

1. Is the manuscript technically sound, and do the data support the conclusions?

Reviewer #1: Yes

Reviewer #2: Partly

2. Has the statistical analysis been performed appropriately and rigorously? 

Reviewer #1: Yes

Reviewer #2: Yes

3. Have the authors made all data underlying the findings in their manuscript fully available?

Reviewer #1: Yes

Reviewer #2: Yes

4. Is the manuscript presented in an intelligible fashion and written in standard English?

Reviewer #1: No

Reviewer #2: Yes

5. Review Comments to the Author

Reviewer #1: Firstly, I would like to thank the authors for preparing this manuscript. This research presents a very intriguing idea with substantial importance in terms of novelty. It contributes to the development of AI-related advances and their applicability in urban design research, advocating for the utility and functionality of such recently developed tools, which are becoming considerably widespread nowadays. Therefore, I would like to suggest the publication of this paper, subject to addressing the following comments:

1. There are several unusual mistakes in the writing (e.g., line 4 of the abstract: repetitive statement of GPT-4o (GPT-4o)). Additionally, the numbering of sections and subsections is incorrect, and the conclusion section does not have a number. Moreover, the citation style is inconsistent throughout the paper; some references use a number-based format, while others follow APA. The authors must ensure consistency in the style of the paper. In general, the fluency and grammatical accuracy of the text need to be thoroughly checked.

2. The Introduction section is too brief. It should be expanded to better contextualize the topic, clearly define the variables associated with this study, and enhance its bibliographic resources.

3. Since the first subsection of the literature review forms the core of the analysis, it would benefit from a wider range of bibliographical resources. This includes elaborating on pedestrian-friendly infrastructure, the five visual elements of walking, the perception of security, street furniture, and urban design features influencing walkable urban spaces. Additionally, addressing the limitations of image semantic segmentation in the last paragraph of the ‘Walkability Perception’ subsection requires supporting evidence.

4. The clarity of Figure 1 must be improved. It appears to have been screenshotted while in modifiable mode, and some arrows are incomplete. Furthermore, each figure must be cited in the text and accompanied by suitable explanations. Likewise, figure captions should be described in detail.

5. The authors need to provide information about the sample size of the conducted survey in the Methods section, along with the rationale behind the chosen number of participants.

6. Demographic data should not be presented in such a chaotic manner. It would be more effective to summarize the participation range of each group using percentages to enhance the fluency of the text.

7. One of the most significant limitations of this study is that some of the indicators of walkability mentioned (e.g., livability and accessibility) may not be measurable using the current tools. This limitation should be explicitly noted in the Conclusion section. Additionally, the suggestions for further research should be improved by elaborating on other potential applications of AI in interpreting, organizing, and analyzing urban planning research.

Reviewer #2: Overall

The manuscript presents an interesting and innovative approach to examining urban walkability through the lens of both GPT-4 and human perceptions. The attempt to bridge AI-driven analysis with human judgment in this domain is both timely and significant. However, the study still requires further refinement and clarification in several areas to improve its overall rigor and readability.

Q1:

The overall quality of the images in the manuscript is suboptimal. There are several areas where the visuals could be improved for better clarity and more precise communication. For example, in Figure 1, the arrows below the image are unclear regarding what they are pointing to or connecting. Additionally, the image appears to have been captured as a screenshot, which is not ideal for a scientific publication. This screenshot retains an active selection box.

Q2

Section 2 Methodology: The description of how images were selected for evaluation is unclear. It is crucial to elaborate on the selection criteria for images and the participant demographics (e.g., age, profession, cultural background) to ensure the generalizability of results.

It would be helpful to mention the number of human participants and the diversity of the urban areas assessed. More specific details on the data (e.g., demographic diversity of participants or add more showcase of cityscape images to better illustrate) would strengthen the findings.

Q3

In the result section, the manuscript presents similarity scores (e.g., 0.4575) without sufficient explanation of how these scores were calculated, their statistical significance, or what constitutes a significant alignment between GPT-4o and human evaluations. The concept of a "coherence score" is not mentioned or defined, which may be critical for understanding the reliability and meaningfulness of the results.

Suggestion:

1. Provide a detailed explanation of how the similarity scores were computed, including the citations, equations, or models used to derive these values.

2. Clarify the significance of the similarity scores in the context of human and AI evaluation alignment. For example, what threshold value of the score represents a significant alignment between GPT-4o and human responses? Are there statistical tests that were performed to validate these scores?

3. If "coherence score" is a part of the analysis, please define it explicitly and explain its role in assessing the results.

Q4

The paper mentions the potential of AI to assist in urban planning tasks but stops short of providing a clear set of recommendations for the practical implementation of AI in urban design. The study could benefit from a deeper discussion of how AI-based assessments of urban environments could influence urban planning. Are there specific applications where GPT-4o might be more useful? What are the limitations for human-cantered decisions?

For example, how do cultural or personal factors influence human judgments of walkability? How might GPT-4o handle these subjective differences?

Q5

There is a minor error of the section serial number in the manuscript, where Section 2 is followed directly by Section 4, without Section 3 being included.

6. PLOS authors have the option to publish the peer review history of their article (what does this mean? ). If published, this will include your full peer review and any attached files.

**Do you want your identity to be public for this peer review?** For information about this choice, including consent withdrawal, please see our Privacy Policy .

Reviewer #1: No

Reviewer #2: **Yes: ** Waishan Qiu

---

## [Author Response · Author response to Decision Letter 1]

5 Feb 2025

Thanks for your comments and suggestions making this manuscript much better.

---

## [Editor Report · Decision Letter 1]

18 Mar 2025

Urban Walkability Through Different Lenses: A Comparative Study of GPT-4o and Human Perceptions

PONE-D-24-54765R1

Dear Dr. Saeidi-Rizi,

We’re pleased to inform you that your manuscript has been judged scientifically suitable for publication and will be formally accepted for publication once it meets all outstanding technical requirements.

Kind regards,

Abeer Elshater

Academic Editor

PLOS ONE
---

## [Editor Report · Acceptance letter]

PONE-D-24-54765R1

PLOS ONE

Dear Dr. Saeidi-Rizi,

I'm pleased to inform you that your manuscript has been deemed suitable for publication in PLOS ONE. Congratulations! Your manuscript is now being handed over to our production team.

Kind regards,

on behalf of

Prof. Abeer Elshater

Academic Editor

PLOS ONE